

# Addiction peer recovery coach training pilot: assessment of confidence levels

Nicholas Guenzel[1] and Hongying Dai[2]

[1] University of Nebraska Medical Center, Lincoln, NE, USA
[2] University of Nebraska Medical Center, Omaha, NE, USA

## ABSTRACT

**Background:** Peer recovery coaches (PRCs) have become a critical tool in addiction treatment in many areas of the world. Despite this fact, no identified research has examined the process or impact of PRC training. Furthermore, no scales were identified to measure trainee confidence in various PRC techniques. The goal of this article is to analyze the process and immediate impact of PRC training of twelve American Indians (AIs) in a culturally-specific program. We focus most specifically on trainee confidence levels.

**Methods:** No written consent was obtained and completion of the assessment was considered consent. Trainees completed self-assessments before and after the training. The self-assessment examined nine areas ranging from understanding the role of PRCs to knowledge of effective PRC techniques. Paired *t*-tests were used to assess for changes in individual trainee responses between the pre- and post-assessments.

**Results:** Pre-training responses ranged from moderate to high. Questions with the lowest average confidence levels address PRC activities or specific techniques to facilitate recovery. All nine questions showed statistically significant mean improvements in the post-training self-assessments. Questions regarding specific PRC activities and techniques showed the greatest improvement. Questions relating to helping people more generally showed the smallest improvement. Average post-training responses fell within a very narrow range indicating relatively consistent confidence levels across skills. Analysis indicates participants were possibly over-confident in certain areas (i.e., maintaining boundaries). This small pilot represents an initial attempt to measure confidence levels of PRC trainees. The findings may inform future training by identifying certain areas where emphasis might be most helpful for trainees. In addition, it is hoped that this work will encourage more systematic analysis of the impact of PRC training on individuals.

# INTRODUCTION

Individuals seeking recovery from addiction have doubtlessly been using peer recovery for centuries. Peer recovery groups such as Alcoholics Anonymous were formalized in the early 20th century (*Kurtz, 2010*). Specific volunteer and paid roles for peer workers have been established in various ways since the 1840s (*White, 2010*). However, early roles almost certainly lacked the preparation and standards seen in peer recovery coach (PRC)

Corresponding author
Nicholas Guenzel,
nguenzel@unmc.edu

certifications of today. Some authors have conceptualized PRC services as a bridge between professional addiction treatment and recovery mutual aid (*White, 2010*). Research has identified that PRCs connect with the people they serve based on "trust and shared lived experiences" rather than the professional training of healthcare providers (*Collins et al., 2019*, p. 6). Although peer recovery coaches hold great promise, limited research has examined their training. The purpose of this article is to assess a peer recovery coach training program for American Indians (AIs) that was conducted in the fall of 2019.

In contrast to many other interventions to address health problems, it seems intuitive for individuals who have succeeded in the process of recovery to guide others seeking the same. In part, it appears this natural process has led to great diversity in peer recovery services. Various titles including peer recovery coaches, peer recovery specialists, peer workers, peer navigators, peer mentors, wellness coaches and others have been used for individuals providing support. Although the use of PRCs has grown internationally, the specific roles of support individuals have also varied greatly (*White, 2010*). In some cases, the services delivered by individuals with these titles may be quite similar but in others they may be very diverse. Without standardized terms and defined roles, it is difficult to compare the services of support individuals across programs.

Interventions have ranged in duration from as short as one meeting with a telephone follow-up to long-term relationships over the course of months or years. Many programs have targeted specific populations including individuals with co-occurring mental health problems, pregnant women, individuals in the criminal justice system, people living in supported housing, and people with HIV (*Reif et al., 2014*). Some researchers have found that the use of PRCs has been associated with reduced substance use and increased medical service engagement (*Cos et al., 2019*). However, the heterogeneity of PRC studies has limited the ability of researchers to use metanalyses to draw clear conclusions (*Eddie et al., 2019*). More standardized training and assessment could play a role in improving consistency across studies and projects (*Elsinger & Rentsch, 2019*). Only recently have researchers started to examine the needs of PRCs including the challenges of vulnerability, authenticity, boundaries, stigma, and lack of recognition (*Miler et al., 2020*). To further work towards more consistent trainee assessment, the first author developed a self-assessment scale measuring trainee confidence in a variety of PRC tasks.

Searches of the literature did not locate PRC treatment programs specific to any cultural groups other than AIs (*Kelley et al., 2017*). Many AI cultures may be particularly well suited for PRC work for several reasons. First, AIs have a long history of "recovery circles" where members in recovery support each others' efforts, often using AI traditions such as sweat lodges and talking circles (*White, 2010*). Second, unlike many European cultures, intoxicating substances appear to have played a very minimal role in AI groups outside of ceremonial use (*Frank, Moore & Ames, 2000*). Third, the AI values of respect for traditional ways and a more collective orientation may lead AIs to be more receptive to PRC interventions (*Rowan et al., 2014*).

Most states have established a PRC certification process. No national process has been established for PRC certification and no standard competencies have been proposed.

The number of hours of specific PRC education varies by state but are commonly between 60 and 100 hours. Most states require paid or volunteer experience which varies greatly between 100 and 1,000 hours (*Blash, Chan & Chapman, 2015*). In addition, many certifications require a defined number of hours of supervised work experience that can vary greatly between 24 and 500 hours. Lastly, aspiring PRCs are required to pass a certification exam in most states (*Myrick & Del Vecchio, 2016*). The broad requirements for certification make it clear that a great amount of diversity exists in PRC preparation. No articles reviewing the PRC training process were located so this work will seek to examine the ways in which one training program prepared PRCs.

The training examined in this article was part of a study to assess the impact of AI PRCs in assisting other AIs in the process of recovery from alcohol addiction. The project is being carried out in collaboration with the Nebraska Urban Indian Health Coalition, the Society of Care, and the Southeast Nebraska Native American Collation. The training and study incorporate AI values and traditions which include a cultural history of sobriety, spiritual implications for intoxication, and greater consciousness of individual actions on the collective community. In addition, the training and coaching makes use of AI spiritual practices (i.e., sweat lodges, talking circles) which require and can facilitate sobriety. In the understanding of the authors, the AI aspects of the training and coaching give individual coaches more tools to use with individuals in recovery. However, in most cases they do not fundamentally change the role of the coach.

The first author of this article contracted with two trainers from Michigan to provide a 20-hour PRC training program. Six AIs from Lincoln and six AIs from Omaha completed the training over three days in a Nebraska state park. The trainees include four men and eight women ranging in age from 33 to 61.

### Aims

1. Examine the absolute confidence levels in the nine areas addressed by the pre-training self-assessment tool and the relative confidence levels between the areas
2. Examine the change scores between the pre- and post-training assessments
3. Examine correlations between the change scores of the questions

## METHODS

The University of Nebraska Medical Center approved this study (595-19-EP) Trainees were identified through the primary investigator's contacts from previous projects, the Nebraska Urban Indian Health Coalition, the Society of Care, and the Southeast Nebraska Native American Coalition. Please see Fig. 1 for a flowchart of the project.

Trainers used several methods to deliver content. Some information was relayed in the form of presentations. Discussions were also held to facilitate understanding. Numerous videos were shown including autobiographical stories in which AIs reviewed their recovery journeys. The most interactive portion of the training consisted of numerous role-playing activities in which one trainee acted as the PRC and the other as the individual in recovery. The role-playing allowed trainees to practice and demonstrate what
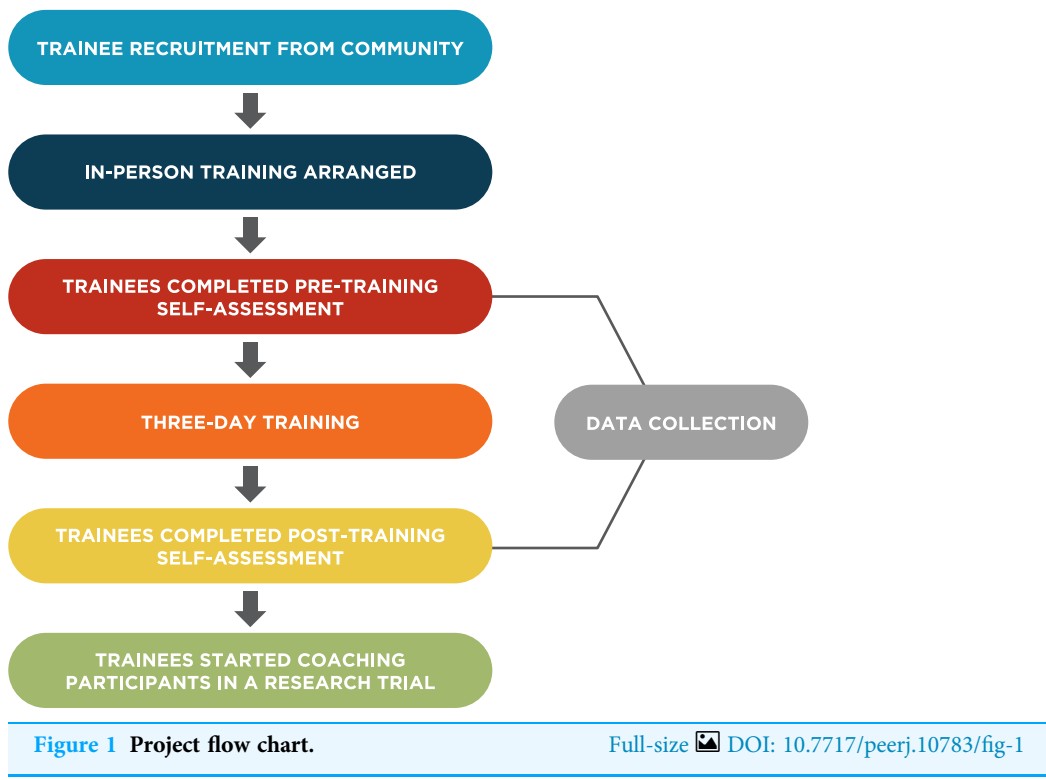

**Figure 1  Project flow chart.**               

they had learned. These exercises also triggered extensive discussion about a wide variety of topics relating to the work of PRCs and addiction more generally.

The program was opened with an elder performing prayers and smudging. The trainers then proceeded with an overview of the program followed by example descriptions of what AI PRCs do in their work. During this time, each participant was asked to describe their background and what brought them to this training. The Anishnaabek Healing Circle was then examined as a model for tribal/urban PRC programs (*Inter-Tribal Council of Michigan, 2020*). The groups were then divided between people from Omaha and those from Lincoln. The two groups were asked to identify the legal, cultural, physical, spiritual, and mental resources in each city. Copies of these lists were then distributed to the trainees for use in assisting the people with whom they work. The next section of the training gave an overview of the structure and function recovery management plans. Trainers then discussed the process of developing recovery management plans.

During the next section, the trainers went into greater detail regarding the roles and responsibilities of a PRC. Specifically, discussion of how PRCs differ from counselors, sponsors, and others helped trainees understand how PRCs operate. Strategies for managing professional boundaries were also discussed including ethics, confidentiality, power differentials, and challenges during crisis management. The trainers then discussed the support available for PRCs and how PRCs can do their part in maintaining effective relationships with their supervisors.

On the final day, the trainers reviewed the additional steps the trainees would need to complete to become certified. This included strategies self-study using the Rhode Island
**Table 1 Pre- and post-training confidence assessment questions.[a]**

|  | Pre: mean (SD) | Post: mean (SD) | Mean change (SD) | p-value[b] |
|---|---|---|---|---|
| 1. I understand the role of a peer recovery coach | 5.4 (2.9) | 8.7 (0.7) | 3.3 (2.7) | 0.0016** |
| 2. I am comfortable in assisting people in an early stage of recovery | 7.3 (1.9) | 8.8 (0.5) | 1.5 (1.7) | 0.0121* |
| 3. I can define the roles and responsibilities of a peer recovery coach | 4.7 (3.0) | 8.6 (0.8) | 3.9 (3.0) | 0.0008*** |
| 4. I am able to help individuals in recovery from addiction | 7.3 (2.0) | 8.6 (0.8) | 1.3 (1.9) | 0.0404* |
| 5. I am skilled at maintaining professional boundaries in challenging situations | 7.1 (1.7) | 8.5 (0.8) | 1.4 (1.3) | 0.0033** |
| 6. I feel self-assured in helping people in recovery | 7.4 (1.6) | 8.7 (0.5) | 1.3 (1.4) | 0.0112* |
| 7. I would do well in helping someone develop a recovery plan | 7.0 (2.1) | 8.6 (0.7) | 1.6 (1.8) | 0.0122* |
| 8. I am able to serve as an effective peer recovery coach | 6.3 (2.4) | 8.7 (0.7) | 2.3 (2.3) | 0.0045** |
| 9. I have effective techniques for helping individuals in recovery | 6.3 (2.5) | 8.4 (0.8) | 2.1 (2.2) | 0.0064** |

Notes:
[a] Each question was rated by a scale from 1 (disagree) to 9 (agree).
[b] Paired $t$-test was performed to compare confidence level before and after training.
* $p < 0.05$.
** $p < 0.01$.
*** $p < 0.001$.

Exam Study Guide, the examination, required work hours, and required supervised PRC activities (*Department of Behavioral Healthcare Developmental Disabilities & Hospitals, 2016*; *Nebraska Department of Health & Human Services, 2020*). The next part of the day was spent exploring and reinforcing skills learned during the previous two days through various role-playing activities. The training concluded with a large group sharing exercise where everyone discussed their thoughts and feelings as they progressed through the training.

The first author developed nine questions based on the specific curriculum of the training to assist in assessing the trainees' confidence in PRC activities (see Table 1). Trainees could circle any number from 1 ("disagree") to 9 ("agree") to indicate their response to the question. Paper copies of the questions were given to participants before the training started. The same procedure was repeated at the end of the training. The authors then entered the pre- and post-responses for each individual into SPSS. Paired $t$-tests were then run to determine if trainee responses changed after the training. Pearson correlations were calculated to measure the association of change in confidence level between each pair of questions. Statistical analyses were performed using SAS 9.4 (Cary, NC) and statistical significance was claimed with $p$-value $< 0.05$.

## RESULTS

### Aim 1

Trainees rated their confidence as fairly high in most measures of the pre-training assessment with means ranging from 4.7 to 7.4 (see Table 1). In five of the seven questions, the mean pre-response was seven or above. Only two questions had mean pre-responses of 5.4 or lower reflecting a moderate level of confidence. However, it is worth noting that the three questions with the lowest level of initial agreement specifically addressed PRC roles and activities. The other question that rated somewhat lower addressed specific techniques to facilitate recovery. All other questions had an average initial response of seven or higher.

**Table 2 Pearson Correlation for the change of confidence levels.[a]**

| Pearson Correlation (p-value) | Assisting | Define | Help | Boundaries | Self-assured | Recovery plan | Effective | Techniques |
|---|---|---|---|---|---|---|---|---|
| **Role** | 0.15 (0.65) | 0.88 (0.0002)*** | 0.24 (0.45) | 0.38 (0.22) | 0.17 (0.59) | 0.04 (0.90) | 0.52 (0.08) | 0.12 (0.71) |
| **Assisting** | | 0.35 (0.27) | 0.72 (0.009)** | 0.66 (0.02)* | 0.76 (0.004)** | 0.73 (0.007)** | 0.76 (0.004)** | 0.82 (0.001)** |
| **Define** | | | 0.33 (0.29) | 0.34 (0.28) | 0.29 (0.37) | 0.04 (0.09) | 0.56 (0.06) | 0.29 (0.37) |
| **Help** | | | | 0.81 (0.002)** | 0.73 (0.007)** | 0.7 (0.02)* | 0.54 (0.07) | 0.81 (0.001)** |
| **Boundaries** | | | | | 0.72 (0.008)** | 0.8 (0.002)** | 0.77 (0.003)** | 0.82 (0.001)** |
| **Self-assured** | | | | | | 0.78 (0.003)** | 0.65 (0.02)* | 0.88 (0.0001)*** |
| **Recovery plan** | | | | | | | 0.54 (0.07) | 0.82 (0.001)** |
| **Effective** | | | | | | | | 0.70 (0.01)* |

Notes:
[a] For each question, change = post-training rating—pre-training rating. Pearson correlation (p-value) are reported in the table.
* $p < 0.05$.
** $p < 0.01$.
*** $p < 0.001$.

### Aim 2

Despite the moderate to high level of initial confidence trainees had in response to most questions and the small sample size, statistically-significant increases in their confidence were found in responses to all nine questions at the $p < 0.05$ level. The standard deviations were also smaller in the post-training assessment relative to the pre-training assessment. As might have been predicted, the largest mean difference between the two surveys specifically asked about aspects of PRCs (understanding the role, defining the role and responsibilities, confidence in serving as an effective PRC, and having effective techniques to help individuals in recovery) and also had the lowest initial ratings. The five questions with the smallest changes (1.25–1.58) related to more general aspects of recovery support rather than PRC-specific skills and tasks. These included being able to help, feeling self-assured in helping, maintaining boundaries, feeling comfortable in assisting people, and helping develop a recovery plan.

### Aim 3

Pearson correlations were calculated to measure the association of change between questions (see Table 2). The negative sign stands for the inverse association while the magnitude of correlation measures the strength of the association. Many strong associations were noted between the questions. The strongest correlations both had a coefficient of 0.88. The first was between "understanding the role of a PRC" and "defining the roles and responsibilities of a PRC." The second was between "feeling self-assured in helping people in recovery" and "having effective techniques for helping individuals in recovery."

### DISCUSSION

Although PRCs are widely employed in helping individuals recover from addiction, very little research has examined the effect of their support and no identified research has focused on their training. To our knowledge, this article represents the first analysis of initial confidence levels of PRC trainees in specific PRC-related areas and the changes that

occurred as a result of the training. The authors contend that systematic analysis of the impact of training methods is essential for the most effective preparation of peer recovery coaches.

A number of previously-undiscussed issues became apparent during the analysis of the self-assessments. For example, it appears likely that trainees were over-confident regarding their ability to maintain professional boundaries as a PRC. This was reflected during the training where trainees made a number of statements reflecting their previous lack of understanding. The initial mean score of 7.08 is especially high for two reasons. First, as indicated by the three questions addressing aspects of PRC work which were scored the lowest in the pre-assessment, trainees admitted to having low information on PRCs relative to more general aspects of recovery. It seems unlikely that individuals who have a limited understanding of PRC roles and responsibilities would be highly skilled in maintaining boundaries which differ significantly from other roles they may have experienced. Second, the question of being skilled at maintaining boundaries necessitates that they not only know what boundaries exist but having techniques needed to enforce them in challenging situations. The responses to the boundaries question and the questions relating to PRCs specifically indicate that trainers need to focus a significant amount of time on defining PRC boundaries. PRCs have roles that are quite different from sponsors, therapists, and others with which trainees may be familiar. Exercises such as role-playing can help trainees practice asserting these boundaries.

Another significant point in this analysis is that there was little change in scores reported that regarded more general aspects of helping individuals in recovery. The most prominent examples included the questions of being able to help people in recovery and feeling self-assured in helping people. It appears that trainees came into the program already having a significant amount of confidence in their ability to assist people in the process of recovery. This confidence is likely beneficial as individuals who believe they can help others will likely be more willing to engage with individuals in recovery than those who are less confident. However, we must note that confidence may not equate to competence. As a result, general principles of assisting individuals in recovery are still a critical part of PRC training.

The last significant point to note is that the greatest increase in confidence was seen in the three questions specifically related to PRC roles. It is logical that these three questions had the lowest initial scores as aspects of PRC work are specific to that role rather than more general principles of recovery support. Individual experiences in treatment, support groups, or other settings foster general recovery knowledge but would be unlikely to give individual insight into PRC work. The low initial scores clearly provided the most room for increased confidence. However, the fact that the post- scores were within a very narrow range with the rest of the questions indicate that the training was effective in at least increasing the confidence of trainees in PRC roles and responsibilities.

This analysis had a number of limitations. First, the analysis is based on a small sample size due, in part, to the fact that such intimate training sessions commonly involve a small number of trainees. Multiple cohorts would likely be needed to draw more reliable conclusions. Second, since the assessments were completed for the training rather than

research, we did not have access to more detailed demographic information for each trainee. The training involved a number of AI values and traditions but the assessment questions did not focus on these specifically so it appears likely that similar patterns may be found in more general populations.

## CONCLUSIONS

The lack of a national certification process or universally-accepted standards should not be a barrier to establishing program were PRCs can help individuals in the process of recovery. This analysis has demonstrated that training can significantly improve the confidence of future PRCs in a wide variety of areas. However, due to the limitations of this analysis, further study and replication is warranted. In particular, more extensive training might start with trainees' baseline level of knowledge which may give insight into confidence and potential misunderstandings in certain areas.

Peer recovery coaches commonly bring valuable experience of addiction but often lack an understanding of the more intricate roles, responsibilities, and techniques of a PRC. Training helps individuals transform this vast experience into tools that will assist others in the recovery from addiction.

### Funding
This work was supported by the University of Nebraska Medical Center. The funders had no role in study design, data collection and analysis, decision to publish, or preparation of the manuscript.

### Grant Disclosures
The following grant information was disclosed by the authors:
University of Nebraska Medical Center.

### Competing Interests
The authors declare that they have no competing interests.

### Author Contributions
- Nicholas Guenzel conceived and designed the experiments, performed the experiments, analyzed the data, prepared figures and/or tables, authored or reviewed drafts of the paper, and approved the final draft.
- Hongying Dai conceived and designed the experiments, analyzed the data, prepared figures and/or tables, authored or reviewed drafts of the paper, and approved the final draft.

### Human Ethics
The following information was supplied relating to ethical approvals (i.e., approving body and any reference numbers):

University of Nebraska Medical Center approved this study (595-19-EP).

## Data Availability

Raw data is available as a Supplemental File.

## Supplemental Information

Supplemental information for this article can be found online at http://dx.doi.org/10.7717/peerj.10783#supplemental-information.

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
