# Peer review of "Addiction peer recovery coach training pilot: assessment of confidence levels"

_PeerJ, doi:10.7717/peerj.10783_

## Round 0.1 · original submission · Minor Revisions

Thank you for your submission. The reviewers have a few minor revisions that you should address.

·

Basic reporting

Overall, the study assesses the peer recovery coach training program for AIs. Well presented paper with clear contribution. writing is clear, professional English language used throughout. Intro & background show context. Literature is well referenced and relevant. The structure conforms to PeerJ standards.

It would help if the text is revised, e.g. the word pregnant women repeated twice in lines 62 and 63. Some terminologies might seem jargon to some readers, e.g. AI. If you mean American Indian, introduce that once, then use the abbreviation. The Anishnaabeck Tribal Umbrella is another example.

Experimental design

Original and primary research within the scope of the journal. Research question well defined, relevant & meaningful. It is stated how the study fills an identified knowledge gap. Good investigation performed to an excellent technical & ethical standard. Methods described with sufficient detail & information to replicate.

However, basic demographic information is essential. I also suggest improving this section by adding a table or a diagram outlining the steps taken, the purpose, duration, materials used in each session, (i.e. study protocol in general). Indicate the sampling technique used and how participants were selected. Perhaps some information about the duration of the training, the structure of the sessions (etc.) would be useful.

I understand that a small sample size would result in a higher margin of error and hence unreliable. But I believe that the study is providing proof of concept rather than statistically significant findings that can be generalized.

Validity of the findings

Novelty is not extensively assessed. All underlying data have been provided; they are robust, statistically sound & controlled. Yet meaningful replication is not encouraged where rationale & benefit to literature is not clearly stated.

It would be great to collect some insight into why confidence was rated high in the pre-training assessment. Such qualitative data would be significant to provide more explanation of the WHY questions. For example, in line 198, it would be useful to know why there was little change in scores reported on the general aspects.

Additional comments

- In line 76, I'm interested to know some of these traditions (juts 2-3 examples) to show how such cultural aspects may have an influence.

- The different labels of the program (lines 56 and 57) were presented as if they are synonyms while there might be subtle differences in terms of what each is trying to emphasize. As a suggestion, C.E. Dickerson work on the logical and scientific foundation for system concepts, principles, and terminology (2008) can be a good approach.

- All techniques need to be cited, such as the Rhode Island Exam Study Guide.

- Study limitations and future directions should be recognized and acknowledged well.

·

Basic reporting

This manuscript meets the majority of the criteria for basic reporting. However, it is a little confusing that the cultural and population focus on this work is not mentioned in the abstract and the first reference at end of first paragraph of introduction just says....for "AIs" .. and it would be helpful to clarify the context for this analysis/study.

Experimental design

The authors are able to meet their objectives with their design of pre and post self assessments; however, the authors do not include any discussion of limitations with this study. For example, the training methods were very culturally specific - how would the results be applied more broadly? Were there any additional factors that might impact the self-assessment process?

Validity of the findings

Despite the very small numbers, the study helps to identify some important concepts and issues to be addressed as the addiction treatment field expands the peer workforce. For that reason, it is worthy of consideration. That training improves confidence levels is not novel, though some of the high pre -training confidence levels are.
Description of the assessment tool is sufficient; conclusions are linked to the data provided.

Additional comments

While this is a very small study, the topic and insights regarding confidence levels can help contribute to the expanding work on training peers. The abstract and initial section of introduction do not clearly make the link between the general study and the culturally specific aspects of the training. It would be interesting to know your assessment of how culture impacts training.

It would be also be helpful to add your assessment of any limitations of the study.

---

## Round 0.2 · accepted · Accept

Thank you for your revised submission. You have addressed the issues raised by both reviewers to my satisfaction and therefore I am pleased to recommend your paper for publication.